# Serum microRNA Profiles and Pathways in Hepatitis B-Associated Hepatocellular Carcinoma: A South African Study

**DOI:** 10.3390/ijms25020975

**Published:** 2024-01-12

**Authors:** Kurt Sartorius, Benn Sartorius, Cheryl Winkler, Anil Chuturgoon, Tsai-Wei Shen, Yongmei Zhao, Ping An

**Affiliations:** 1Faculty of Commerce, Law and Management, University of the Witwatersrand, Johannesburg 2001, South Africa; 2School of Laboratory Medicine and Molecular Sciences, University of Kwazulu-Natal, Durban 4041, South Africa; chutur@ukzn.ac.za; 3Africa Hepatopancreatobiliary Cancer Consortium (AHPBCC), Mayo Clinic, Jacksonville, FL 32224, USA; 4School of Public Health, University of Queensland, Brisbane, QLD 4102, Australia; 5Centre for Cancer Research, Basic Research Laboratory, National Cancer Institute, Frederick Natifol Laboratory for Cancer Research, National Institute of Health, Frederick, MD 21701, USA; 6CCR-SF Bioinformatics Group, Frederick National Laboratory for Cancer Research, Frederick, MD 21701, USA

**Keywords:** microRNA (miRNA), dysregulated, modulate, gene targets, pathways, hepatitis B-associated carcinoma (HBV-HCC)

## Abstract

The incidence and mortality of hepatocellular carcinoma (HCC) in Sub-Saharan Africa is projected to increase sharply by 2040 against a backdrop of limited diagnostic and therapeutic options. Two large South African-based case control studies have developed a serum-based miRNome for Hepatitis B-associated hepatocellular carcinoma (HBV-HCC), as well as identifying their gene targets and pathways. Using a combination of RNA sequencing, differential analysis and filters including a unique molecular index count (UMI) ≥ 10 and log fold change (LFC) range > 2: <−0.5 (*p* < 0.05), 91 dysregulated miRNAs were characterized including 30 that were upregulated and 61 were downregulated. KEGG analysis, a literature review and other bioinformatic tools identified the targeted genes and HBV-HCC pathways of the top 10 most dysregulated miRNAs. The results, which are based on differentiating miRNA expression of cases versus controls, also develop a serum-based miRNA diagnostic panel that indicates 95.9% sensitivity, 91.0% specificity and a Youden Index of 0.869. In conclusion, the results develop a comprehensive African HBV-HCC miRNome that potentially can contribute to RNA-based diagnostic and therapeutic options.

## 1. Introduction

Hepatocellular carcinoma (HCC) remains an intractable global problem against a background of rising incidence and changing etiology [1]. In 2020, liver cancer mortality was ranked in the top three across 46 countries, with a prediction that 1.3 million people would die from this disease by 2040, indicating a 56.4% increase from 2020. In Sub-Saharan Africa (SSA), 38,600 new cases of liver cancer were estimated, with west Africa as the leading region (17,600/ASR 8.4), followed by east Africa (12,300/ASR 5.0), middle Africa (6 100/ASR 6.1) and southern Africa (2600/ASR 4.6) [2]. Viral infection with hepatitis B remains a primary risk factor in this region [3,4,5] and chronic hepatitis B virus (HBV) infection in SSA has been etiologically implicated in 43% to 80% of total HCC incidence in this region (1,2,3). High HBsAg seroprevalence (>5%) levels also currently persist in multiple SSA countries and South Africa, in particular [6,7].

MicroRNA (miRNA) are often regarded as ancillary epigenetic regulators that act as post-transcriptional gene silencers [8]. These miRNAs collectively repress target mRNA expression in order to ensure homeostasis, and their dynamic fluctuating role is constantly changing as a result of gene transcription and environmental fluctuations [9]. In the HBV-HCC continuum, from asymptomatic HBV infection to the onset of HBV-HCC pathogenesis, an increasing number of miRNAs become dysregulated due to viral infection, epigenetic changes [10], inflammation [11], fibrosis [12], cirrhosis [13] and finally, the onset of HCC. In particular, the HBx protein modulates miRNA expression to influence both its own, as well as its host’s, genome expression in all of the HCC cancer and immune pathways [14].

Despite the early promise of RNA therapeutics and diagnostics as a magic bullet to diagnose early-stage cancer, as well as to modulate aberrant signaling in cancer, this field remains a work-in-progress with regard to their diagnostic and therapeutic potential [15]. Although RNA-based detection of disease is highly predictive, this has not translated into any commercial application, and RNA inhibition-based therapeutics (RNAi) have been confronted with RNA delivery problems and host immunogenic issues [15]. These problems are illustrated by a lack of successful RNAi clinical trials and FDA-approved drugs for the treatment of HCC [16]. Nevertheless, RNAi offers great potential as a therapeutic option for cancer and RNA-based therapeutics are now a reality for the prevention of viral diseases (COVID-19). This paper presents a comprehensive profile of HCC serum miRNA and their gene targets in a South African case study involving high levels of hepatitis B infection and, to the best of our knowledge, is the first comprehensive serum-based mIRNome study on HCC in Africa. Using a sample selected from two case control studies, our specific objectives were to (1) classify significantly dysregulated mappable miRNA, (2) identify their targets, (3) identify the HCC pathways that they modulate and (4) investigate potential miRNA diagnostic panels.

## 2. Results

The results focus on the top 10 upregulated miRNAs and their validated targets (Table 1) and the top 10 downregulated miRNAs and their validated targets (Table 2) but specifically provide supplementary data with respect to all significantly dysregulated miRNAs above a threshold of a unique molecular index (UMI) ≥ 1, LFC > 2 and log fold change (LFC) < −0.5 (*p* < 0.05) (Appendix A) and 91 dysregulated miRNAs and their validated targets above a threshold of unique molecular index (UMI) ≥ 10, LFC > 2 and log fold change (LFC) < −0.5 (*p* < 0.05) (Appendix A). The results then separately discuss significantly dysregulated miRNAs in terms of log fold change before illustrating validated KEGG HCC pathways of the top 10 upregulated and downregulated miRNAs in terms of mean UMI (Figure 1 and Figure 2). Finally, the results present selected miRNA panels as diagnostic agents before selecting and presenting the optimum panel based on its sensitivity and specificity, as well as a heatmap of its associations with a range of disease and cellular processes (Table 3, Figure 3 and Figure 4).

### 2.1. Top 10 Dysregulated miRNAs Exceeding an UMI ≥ 10

The top 10 upregulated miRNAs with a mean UMI ≥ 10 (see Table 1) ranged between a mean UMI expression of 344 (miR-130b) and 65596 (miR-122-5p), and a mean LFC ranged from 2.018 (miR-320b) to 5.142 (miR-4532). Validated targets have only been included in this section to enhance the reliability of the results but are further developed in the discussion. Validated HCC targets of miR-122-5p, the most widely expressed liver miRNA, include β-CAT/ CCNG1/53 /HNF4α and BCL-W. The second highest upregulated miRNA, miR-483-5p, modulated expression of PPARα/TIMP2 and CDK15 (see Table 1).

The top 10 downregulated miRNAs ranged from a mean UMI of 2691 (miR-191-5p LFC −0.569) to a mean UMI of 57609 (miR-16-5p LFC −0.54). Although the most downregulated miRNAs in terms of absolute UMI expression was miR-16-5p, the let-7-5p family members (a-b-f-i), collectively, were the most downregulated miRNAs in terms of both FC (−0.922 to −1.236) and UMI (9798–17716). Interestingly, other significantly downregulated Let-7-5p members (d/e/g) indicated no direct validated target genes. The most dysregulated miRNAs in terms of both UMI and LFC was miR-483-3p (UMI = 44961; LFC 4.89), suggesting that it could potentially play a major modulating role in HBV-HCC pathogenesis.

In terms of LFC and UMI ≤ 10, significantly upregulated miRNAs not already reported in Table 1 were miR-373-3p (UMI = 6.0; LFC 8.422), miR-1290 (UMI = 9.0; LFC 6.375) and miR-4492 (UMI = 5.0; LFC 5.521). Two of these miRNAs in previous HCC studies demonstrated mIR-373-3p targeting TFAP4/CDH1/CSDC2 and PPP6C [72,73,74,75], and miR-1290 targeting SMEK1/FOXC1/SLU7/GLIPR1. The 10 top downregulated miRNAs in terms of LFC ranged between miR-1306-3p (UMI = 2.0; LFC −2.705) modulating FBXL5 [76] and miR-1275 (UMI = 8.0 LFC −2.007) targeting IGFBP1/2/3/E1F5A2 [77,78,79,80,81] and, in general, were less referenced, except for mIR-491-5p (UMI = 3.0; LFC −2.488) targeting PKM2 and SEC61A1/EGFR [82,83,84] and miR-221-5p (UMI = 2.0; LFC −2.146) targeting ITGB5/GCDH and CD44-TGF-B1 [85,86,87,88] (see Appendix A).

### 2.2. Downregulated miRNA Pathways

KEGG pathway analysis (mirPATHv4), which only highlights validated gene targets in specific pathways, indicated that the 10 most downregulated miRNAs (mIR-191-5p, miR-26b-5p, miR-146a-5p, miR-142-3p, miR-126-3p, let-7i-5p, let-7f-5p, let-7a-5p, let-7b-5p and miR-16-5p) indicate viral oncogenesis as the top ranked association. This panel can influence HBx-induced HCC pathogenesis via modulating multiple pathways (see Figure 1). In the HBx dysregulated HBV-HCC pathways, this panel of MIRNAs can influence a number of genes in the TGF-B/SMAD pathway by targeting validated gene targets like TGFBR1/SMAD3/, CDKN1A and MYC to influence both oncogenesis and the loss of tumor suppression. In the P13K/AKT pathway, the top 10 downregulated miRNAs target PIK3A/AKT2/CHUK to influence NF-kB1 promotion of BCL2 to influence invasion, metastasis and anti-apoptosis, and in the JAK/STAT pathway, this miRNA panel targets STAT1/3 to influence hepatocyte proliferation. MAPK signaling is targeted by the top 10 downregulated miRNAs by influencing the expression of VDAC3/SRC/GRB2/SOS and KRAS to influence MAP3K, MAP3/JUN and SMAD4 to influence oncogenesis in the SAPK/JNK. Similarly, MAPK signaling is influenced in the RAS/RAF/ERK pathway by this panel of miRNA targeting ARAF/MAP2KT/MAPK1/MYC and STAT3 to promote cell proliferation. Validated targets in cell cycle controls that are also dysregulated by the HBx protein include TP53/FAS, CDKN1A/CDK2, CCNE1/RBI and E2F1 to influence the loss of cell cycle controls. Our Table 2, which only provides some validated gene targets, does not list many of the validated gene targets highlighted by miRPATHv4 KEGG analysis. It is important to stress that downregulated miRNAs mostly promote oncogenesis by failing to regulate oncoprotein expression (e.g., MYC) but can also moderate oncogenesis by reduced modulation of tumor suppressor expression (e.g., TP53).

The miRPATHv4 results further indicate that the complexity of the interaction of these top 10 downregulated miRNAs is emphasized by their additional role of promoting immune evasion in the innate immune system by targeting the MyD88 dependent and independent pathways. This panel also modulates innate immune evasion by modulating the MYD88 pathway by targeting MYD88/TIRAP/IRAK1/TAB2/MAPK3K7/CHUK/MAPK2K1/MAPK2K3/MAPK2K4/MAPK1/MAPK14/MAPK8/JUN/TLR4/TICAM/MAVS and TRAF3 /TBK1/IRF3 to influence immune evasion.

### 2.3. Upregulated miRNA Pathways

KEGG bioinformatic analysis (mirPATHv4), which identifies validated targets and pathways in HCC, also indicates that the top 10 upregulated miRNAs (miR-130b-5p, miR-320d, miR-483-3p, miR-1246, miR-320b, miR-192-5p, miR-4532, miR-320c, miR-483-5p and miR-122-5p) also influence expression in multiple HCC pathways (see Figure 2). These miRNAs appear to be less linked with viral carcinogenesis than the top 10 downregulated miRNAs. In the P13K/AKT pathway, this upregulated miRNA panel can target IGF2/IGF1R/PIK3CB/AKT3/RPS6KB to influence survival, while in the p53 signaling pathway, it can target CDK4/CCND1/E2F3 to modulate G1/S progression. In the TGF-B signaling pathway, these miRNAs target SMAD2/4 to influence the loss of inhibitory effect to promote oncogenesis. In the WNT signaling pathway, the miRNA set targets WNT2 and FZD5/LRP6/DVL3/GSK3B /LEF/MYC/CCND1 to influence proliferation and differentiation, and in the MAPK signaling pathway, these miRNAs target PIK3CB/AKT3 and MTOR/RPS6KB to modulate survival. The miRPathv4-identified targets are not necessarily listed in our Table 1; miR-122-5p, for example, has 32 gene targets regarding proteoglycans in cancer and 16 targets regarding TGF-B/SMAD signaling, and our Table 1 only lists 7 targets for this miRNA in HCC studies.

### 2.4. Diagnostic miRNA

The results, indicating that potential diagnostic miRNA panels differentiate cases from controls, showed that five miRNA panels, namely, let-7a-5p and miR-1246, let-7f-5p and miR-4532, let-7a-5p and miR-320c, miR-122-5p and let-7a-5p, and miR-320c/b and let-7a-5p, all developed an AUC > 90% (see Table 3). All five miRNA panels recorded AUCs > 90%, indicating *p*-values < 0.01 and an UMI level > 500 (see Table 2). These miRNA panels included miRNAs with the greatest UMI and LFC that ranged between an UMI of 693 and 65596 and an LFC of 2.018 and 5.142 (upregulated) and an LFC of −1.155 and −1.236 (downregulated).

The specificity and sensitivity of the miRNA panel of let-7a-5p, miR-320b/c indicated the most powerful predictive ability, with 95.9% sensitivity and 91.0% specificity at the cut-off of 0.727 (see Figure 3). The Youden Index for this miRNA panel was 0.869184, confirming high levels of sensitivity and specificity.

A heatmap was then developed for the selected miRNA panel (see Figure 4, miRPATHv4), indicating that let-7a-5p has been significantly associated with multiple types of pathogenesis, cell processes and signaling pathways. Only the Hippo signaling pathway is modulated by all three of the miRNA panels, while let-7a-5p is strongly associated with the modulation of the Adherens Junction, viral carcinogenesis, hepatitis B, thyroid hormone signaling, cell cycle, oocyte meiosis, lysine degradation and ECM–receptor interaction.

Further analysis, using miRTargetLink2.0 and miRPATHv4, illustrates 64 strongly validated gene targets (Appendix A), with hsa-Let-7a-5p targeting 45 genes (<0.001), hsa-miR-320b targeting 10 genes (<0.001) and has-miR-320c targeting 9 genes (<0.001), respectively. Further analysis of the selected diagnostic panel indicates that it strongly targets DICER1, CASP3, MYC, CDK6, NRAS, EZH2, TRIM71, HMGA2, EGFR, STAT3, HRAS and KRAS (https://mpd.bioinf.uni-sb.de/ accessed on 10 October 2023). KEGG pathway analysis indicates that the selected miRNA panel can specifically target SMAD3/CDKN1A/MYC in the TGF-B/SMAD pathway to influence tumor suppression and oncogenesis, as well as target CASP3 in the P13K/AKT pathway to influence apoptosis. This panel also modulates TP53 signaling by targeting CDKN1A, as well as STAT3 in the JAK/STAT pathway to influence proliferation in both of these pathways. In the MAPK pathway, this panel can also influence proliferation by modulating NRAS and MYC/STAT3 expression (https://diana-lab.e-ce.uth.gr/app/miRPathv4 accessed on 10 October 2023) (also see Appendix A and Table 1 and Table 2).

## 3. Discussion

The results present a profile of serum miRNA dysregulation in 98 HBV-HCC cases versus 48 controls, of which 30 miRNAs were significantly upregulated and 61 were significantly downregulated. In this section, we briefly compare these results with similar studies in other settings before discussing their modulatory role in the HBV-HCC pathogenesis continuum, specifically examining their role in inflammation, fibrogenesis, carcinogenesis, angiogenesis, apoptosis, HCC pathogenesis, metastasis and migration–invasion, as well as making specific reference to their epigenetic and immune-modulatory functions. Finally, this section discusses the potential role of serum-based miRNAs as diagnostic agents.

Comparative studies: In a comparative serum miRNome study, 214 miRNA families were listed; the let-7/98/4458/4500, miR-320abcd/4429, miR-378/422a/378bcdefhi and miR-17/17-5p/20ab/20b-5p/93/106ab/427/518a-3p/519d families were the most represented families. Similar to our results, this study indicated that miR-122-5p was highly elevated and miR-320a/b/c was among the top ranked families in HCC patients [27]. In a miRNome study focusing on the most abundantly expressed miRNAs in liver tissue, miR-122-5p, let-7a-5p, let-7b-5p and let-7f-5p were listed in the top five [89], thus corresponding with our results which indicated abundant expression in HCC serum. Interestingly, this study also indicates that miR-199a/b-3p, the 11th most downregulated miRNA in our results, is a potential therapeutic agent because it can target tumor-promoting PAK4 to suppress HCC growth by inhibiting the PAK4/Raf/MEK/ERK pathway. Interestingly, in a study involving dysregulated HCC miRNAs in tumor tissue in the absence of viral infection, 86 miRNAs were identified as significantly dysregulated at a cut-off of log2 (FC) > 1 which included 26 upregulated and 60 downregulated miRNAs [90]. In this study, none of the top 10 upregulated or downregulated miRNAs in terms of LFC coincide with our results.

HBV infection and inflammation: The literature demonstrates that two important dysregulated miRNA family members in our results, the let-7 and miR-122 families, play a significant role in modulating both the expression of the HBV virion in host hepatocytes as well inflammation. A number of studies indicate that let-7a/b/c/d/e/f/g/i are frequently downregulated by the HBx protein in early-stage infection in the PME and therefore can fail to modulate inflammatory proteins like Il-6 and IL-10 [91,92]. Since the literature confirms early-stage miRNA dysregulation, our results, which reflect a range of downregulated let-7-5p family members in (late stage) HBV-HCC serum, suggest that these miRNAs might remain consistently dysregulated throughout the HBV-HCC continuum; however, this needs to be validated by additional studies. Similarly, many studies indicate that miR-122 family members can be upregulated in early-stage pathogenesis [93,94], and both miR-122-5p and miR-151a-3p have been identified as biomarkers for chronic hepatitis B infection (CHB) [95]. Our results, reflecting later stage HCC, indicate that miR-122-5p/3p were highly upregulated, posing the question as to whether this miRNA family remains upregulated in both the PME and tumor microenvironment. Similarly, our results suggest that miR-15a [96,97,98], mIR-17 [91] and miR-182 [99] family members can be downregulated in early HBV infection and in late-stage HCC. Our results also indicate that upregulated miRNAs like miR-192-5p could be upregulated in early-stage HBV infection [100,101], as well as in later stage HBV-HCC serum targeting XIAP/TRIM44, SEMA3A and FABP3/YY1/PABPC4 [29,30]. Conversely, miR-151a-3p, which has been cited as a potential biomarker, can be upregulated in early-stage HBV infection [95], whereas our results indicate that this miRNA is significantly downregulated and targets ATM [102]. Interestingly, the same miRNA can select different targets in different stages of pathogenesis, and downregulated miR-15a, for instance, fails to repress TGF-B/SMAD7 signaling in early infection [96,97,98], whereas our results indicate that downregulated miR-15a-5p can also repress GLI2/IGF1/PD1/eIF4E/BDNF/E2F3 in HBV-HCC serum [103,104,105,106,107]. Our results indicate that miRNA dysregulation is dynamic and that it can remain the same or change from early-stage HBV infection and inflammation to the onset of carcinogenesis, with the literature indicating different validated gene targets along the HBV-HCC continuum.

Fibrogenesis: Hepatic fibrosis is characterized by the accumulation of extracellular matrix (ECM) depositions, changes in the hepatic architecture, scarring and increases in fibrous tissue, liver stiffness and alterations in the vasculature in response to a changing microenvironment [108]. A central event initiating liver fibrogenesis is the activation of hepatic stellate cells (HSCs) [109,110]. Multiple miRNAs that modulate fibrogenesis become dysregulated in this pre-malignant environment (PME) and upregulated miR-122-5p, for example, has been correlated with the degree of fibrogenic damage [95,111,112]. Again, our results suggest that this miRNA, which becomes dysregulated in the fibrogenic PME, can remain dysregulated in HBV-HCC serum. Similarly, our results indicate that upregulated miR-140-5p is also upregulated in cirrhosis [113] and downregulated miR-151a-3p is also downregulated in the PME as a response to fibrogenesis [95]. Interestingly, miR-151a-3p is also cited as a biomarker for liver injury caused by CHB [95]. Conversely, upregulated miR-125a-5p targeting F1H1 in fibrogenesis [114,115] is downregulated in our results, targeting PTPN1/MAP3K11/SIRT7/VDR/MACC1/LASP1 and ErbB3 [115,116,117,118,119,120,121], suggesting that miRNA expression can be differentially manipulated in hepatocarcinogenesis. Similarly, miR-142-3p, which has been reported as upregulated in fibrogenesis in the PME [113], is one of our top 10 downregulated miRNAs targeting LDHA/HMGB1/ RAC1, ZEB1/CD133 and SLC3A2 in HBV-HCC [122,123,124,125,126,127]. The results emphasize the dynamic nature and complexity of miRNA-induced regulation in the HBV-HCC continuum, indicating that many of the same miRNAs may be differentially dysregulated by fibrogenesis in the PME vs. the tumor environment, or alternatively, that they remain consistently dysregulated in the same direction but modulate different targets. Further validation of these statements is required.

HBV-HCC pathogenesis: HBV-HCC pathways typically include aberrant miRNA expression in the RB1-TP53 suppressor networks, the WNT pathway, PI3K/MAPK pathways and the JAK/STAT pathway [128,129]. HBV infection dysregulates a wide range of host gene expressions by initiating deletions, amplifications, mutations and epigenetic changes or by targeting microRNA loci or their transcription factors [130]. Typically, dysregulated miRNAs enhance onco-protein expression or repress tumor suppressor activation in the HCC pathways [131] and therefore play a modulatory role in cell proliferation, apoptosis, angiogenesis, invasion, migration and metastasis, as well as modulation of the immune system and the epigenetic machinery. Although multiple miRNAs modulate HBV-HCC pathogenesis, two ubiquitous liver miRNA families in our results, namely, miR-122 and the let-7 family members, illustrate how miRNAs modulate multiple cellular functions and processes in the HBV-HCC continuum. The miR-122 family modulates proliferation, invasion and apoptosis in HCC pathogenesis by targeting PBF, ADAM10/Cyclin G1 and Igf1R/ADAM 17/BCL-W/NDRG3 [39,40,97,132,133], as well by influencing EMT, cell migration, invasion and metastasis [97,134]. Similarly, the tumor suppressor let-7-5p family members play an important modulatory role in multiple human cancers including HCC by repressing oncogenic targets such as EGFR/LIN28B/ HMGA2 and C-MYC that are all important players in oncogenesis [135,136]. Let-7 family members can also modulate angiogenesis/growth/migration and inflammation, as well as modulate TP53 tumor suppressor function by targeting STAT3/RAS/HMGA2/MYC/IL-6/IL-10/TLR-4/COL1A2 and NGF /BCL-XL [64,92,136,137]. Collectively, our results suggest that 91 significantly dysregulated miRNAs with an UMI ≥ 10 probably play a significant modulatory role in HBV-HCC pathogenesis by influencing cell proliferation, angiogenesis, migration and invasion, and metastasis; however, further validation is suggested to test individual miRNA targets and their effect.

Cell proliferation: KEGG pathway analysis indicated that the dysregulated miRNAs in our results can influence cell proliferation in HBV-HCC pathogenesis. The upregulated tumor suppressor miR-9-3p, for instance, can attenuate cell proliferation by repressing oncogenic ERK1/2, AKT and B-CAT expression [138]. Conversely, upregulated miR-452-5p, which is cited as being consistently elevated in HCC tissue [139], can promote cell proliferation and the progression of HCC by targeting COLEC10 [140]. Downregulated miR-142-5p, on the other hand, fails to repress cell proliferation by its reduced modulation of LDHA that is often aberrantly expressed in HCC [122]. Downregulated miR-125a-5p also fails to inhibit cell proliferation by failing to mediate the PI3K/AKT/mTOR pathway [119] and ERBB3 [121]. Conversely, upregulated miR-483-5p inhibits cell proliferation and fibrosis by targeting PPARα and TIMP2 [34]. Other miRNAs in our results that might modulate cell proliferation in HBV-HCC include miR-874-3p [141], miR-451a [142,143,144,145,146,147], miR-182-5p [148,149,150,151,152,153], miR-98-5p [154,155,156,157,158], miR-363-3p [159,160,161,162] and miR-339-3p [163,164,165].

Apoptosis: Normal apoptotic response is frequently dysregulated in HBV-HCC pathogenesis, and miRNA dysregulation can both enhance as well as attenuate apoptosis. Upregulated miR-494-3p, for instance, represses apoptosis by targeting the PTEN tumor suppressor and promoting PI3K/AKT expression [166], while upregulated miR-494-3p can promote apoptosis by targeting TRAF3 to attenuate hepatic stellate cell activation [167]. Downregulated miR-26-b-5p also attenuates apoptosis by failing to modulate KPNA2, thus contributing to the inactivation of p53 signaling [49]. Similarly, upregulated miR-192-5p promotes survival and suppresses apoptosis by repressing PABPC4 [30]. Our results reflect multiple miRNAs that might influence apoptosis in the HBV-HCC continuum including miR-199a-3p [168,169,170,171,172,173,174,175,176,177,178], miR-584-5p [179,180,181,182], miR-98-5p [154,155,156,157,158], miR-454-3p [183,184,185,186], miR-4516 [187,188] and miR-483-3p [20,21], but further investigation should validate this.

Angiogenesis: Multiple miRNAs can regulate angiogenesis in order to nurture or restrict the expansion of the vasculature microenvironment. Upregulated miR-34a-5p, for instance, can attenuate angiogenesis and cell proliferation by regulating VEGFA expression [189]. Conversely, upregulated miR-210-3p promotes angiogenesis by targeting SMAD4 and STAT6 [190] and downregulated miR-26b-5p fails to repress angiogenesis because of its reduced ability to modulate the expression of E-CAD/SNAIL1/MMP2 [48]. In another study ranking the top 10 downregulated miRNAs in HCC with/without vascular invasion, miR-126-3p (log2FC of −3.36) was an important regulator of ADAM9 expression in HBV-HCC [57,191]. Other miRNAs in our results that might influence angiogenesis in the HBV-HCC continuum include miR-494-3p [166,167,192,193,194], miR-126-3p, miR-146a-5p, miR-199a-3p and miR-491 [195].

Migration and invasion: Multiple miRNAs modulate the expansion of the tumor microenvironment that is underpinned by the migration and invasion of cancer cells. Dysregulated miRNAs in our results indicate that downregulated miR-16-5p, for instance, can fail to repress the invasion and migration of HCC cells by its reduced modulation of IGF1R protein expression [68]. Conversely, upregulated miR-34a-5p suppresses the invasion and metastasis of liver cancer by targeting the transcription factor YY1 to mediate MYCT1 upregulation [196]. Interestingly, hypoxia-related miR-210-5p and miR-210-3p can regulate hypoxia-induced migration and epithelial–mesenchymal transition in hepatoma cells [197]. Conversely, downregulated miR-26b-5p fails to repress epithelial–mesenchymal transition (EMT), migration and invasion by its reduced repression of SMAD1 [47]. Other miRNAs in our results that might influence migration and invasion in HBV-HCC pathogenesis include miR-452-3p [139,140,198,199,200], miR-625-3p [201,202] and miR-15a-5p [103,104,105,106,107].

Metastasis: Multiple miRNAs modulate the ability of HBV-HCC-related metastasis. Downregulated miR-142-3p, for instance, fails to inhibit the metastasis of hepatocellular carcinoma cells by its reduced repression of HMGβ1 gene expression [123]. Similarly, downregulated miR-126-3p also fails to suppress tumor metastasis and angiogenesis of hepatocellular carcinoma by its reduced ability to repress LRP6 and PIK3R2 expression [55]. Downregulated miR-451a also fails to repress metastasis and EMT because of its reduced modulation of YWHAZ and ADAM10 [143,144], and upregulated miR-494-3p can enhance HCC metastasis by targeting BMAL1 [192]. Other miRNAs in our results that might potentially influence metastasis in HCC pathogenesis include miR-96-5p [151,203,204,205], miR-1246 [22,23,24,25,26] and miR-210-3p [190,197,206,207].

miRNA as an ancillary epigenetic system: In the HBV-HCC continuum, the HBx protein can influence epigenetic changes like DNA methylation, histone modifications and Polycomb proteins that can dysregulate a specific subset of miRNAs often labeled as epi-miRNAs [8]. In turn, miRNA expression can modulate downstream epigenetic machinery, suggesting epigenetic feedback loops that involve upstream epigenetic manipulation of miRNA expression that can then target downstream epigenetic targets [97,208,209]. In the HBV-HCC continuum, HBx-dysregulated miRNAs, therefore, are ancillary epigenetic regulators that can be subject to upstream epigenetic modulation. In the HBV-HCC continuum, DNA methylation can downregulate miR-1/-122/-124/-132/-/148/-200/-205 [210,211]. Conversely, histone acetylation or HDAC inhibitors can upregulate miR-224/-29/-155/-17-92, and histone methylation can downregulate Let-7c/miR-101/-125b/-139-5p [97]. Our results reflected in Table 1 and Table 2 suggest that a number of these miRNAs interact with the epigenetic machinery, as well as play other roles in HBV-HCC pathogenesis. For example, miR-140-3p can modulate DNA methyltransferase 1 (DNMT1) which results in changes in NF-kB signaling to promote carcinogenesis in HBV-HCC [212]. Interestingly, it has been widely reported that the HBx protein can directly use epigenetic machinery to promote NF-kB signaling and promote hepatocarcinogenesis [213]. Other examples of upstream epigenetic machinery modulating miRNA expression includes the histone H3 lysine 27 (H3K27) tri-methylating enzyme, an enhancer of Zeste homolog 2 (EZH2) that can be recruited by the HBx protein to downregulate miR-139 family members like miR-139-5p [214] (see Appendix A).

miRNA modulation of immune response: The innate and adaptive immune systems are influenced by an elaborate network of genes whose expression is controlled by extracellular signaling, epigenetic modifiers, transcription and splicing factors, translational protein modifiers and an extensive network of miRNAs [215]. In HBV-HCC pathogenesis, a single miRNA can play multiple roles in both the innate and adaptive immune systems. Two miRNA families in our results that modulate immune response, namely, miR-155 and miR-34a family members, can both be dysregulated by the HBx protein [14]. The miR-155 family is a multifunctional miRNA that plays a crucial role in the modulation of physiological and pathological processes such as hematopoietic lineage differentiation, immunity, inflammation and cancer [216]. This miRNA is expressed in a variety of immune cell types, including B cells, T cells, macrophages, dendritic cells (DCs) and progenitor/stem cell populations. Normally, the miR-155 family is found at low levels in most of these cells types until their activation by immune stimuli, such as antigens, Toll-like receptor (TLR) ligands and inflammatory cytokines, which rapidly increase miR-155 expression [217]. This miRNA, therefore, has an important role in regulating cytokine production and inflammation, as well as in modulating myeloid and lymphoid differentiation [218]. In the immune system, miR-155 is unique in its ability to shape the transcriptome of activated myeloid and lymphoid cells [219]. In virally driven modulation of the immune system, the HBx protein can also repress p53-stimulated miR-34 expression in hepatocytes, leading to the upregulation of a macrophage-derived chemokine (CCL22) that stimulates regulatory T cells (Tregs) which, in turn, represses effector T cell expression, thus allowing HBV expression to increase [97,220]. Upregulated p53-induced miR-34a is also reported to suppress FOXP1, resulting in the inhibition of pro-B cell to pre-B cell transition [221]. Other highly dysregulated miRNAs in our results, like the let-7 family members, play a role in HBV-HCC pathogenesis in the innate immune system [92], and in particular, let-7e and miR-146a can modulate macrophage output, as well as other miRNAs like miR-155 and miR-9 [215]. In addition, miR-146a can modulate Toll-like receptor 4 (TLR4) signaling that has an important role in regulating innate and adaptive immune responses [222]. The most abundantly expressed liver miRNA, miR-122, can also influence immune response by modulating interferon [223], and miR-142-5p can influence T cell response in HCC pathogenesis by modulating PD-L1 expression [224].

miRNAs as diagnostic agents: Our results developed a tentative proposal of alternate miRNA panels that need to be further tested alongside multiple studies that have also profiled diagnostic miRNA panels in HBV-HCC serum. Our results proposed combinations of let-7a-5p and miR-1246, let-7f-5p and miR-4532, miR-320c and let-7a-5p, miR-122-5p and let-7a-5p and the selected panel, namely, miR-320b/c and let-7a-5p, that showed the highest AUC of 0.9822. Of these three miRNAs, only let-7a-5p is supported by multiple studies indicating that let-7a-5p can modulate cell growth, migration and invasion, cell proliferation and apoptosis, and importantly, that it is associated with hepatis B and viral carcinogenesis, as indicated in Figure 4 [60,61,62,63,64]. In a Chinese cohort where 74.8% of patients were HBV positive, three serum-exosome-derived miRNAs (miR-122-5p, let-7d-5p and miR-425-5p) with an AUC of 0.905 were identified as promising biomarkers for identifying HCC patients [225]. In a study in India, let-7a was also listed as a key potential biomarker in a sample where 27% of cases were HBV positive and 24% were HCV positive [226]. Interestingly, let-7d-5p is also listed as a potential biomarker for HCV-HCC and NAFLD [227,228], while miR-425-5p is a well-referenced miRNA in HCC that can promote invasion, metastasis and cell proliferation. This miRNA has also been cited by other studies as a potential diagnostic agent [229,230,231,232]. In our results using miR-122-5p and Let-7a-5p, we developed an ROC of 0.9420, and when we tested for a combination of miR-122-5p, let-7d-5p and miR-425-5p, we developed an ROC of 0.9241, but miR-425-5p was not significant. In other miRNA biomarker studies, miR-122-5p has been listed as a key potential biomarker [19,233]. In a meta study of HBV-HCC miRNA biomarkers combining twenty-three studies from China and two from India, miR-125b emerged as the most promising biomarker for HBV-HCC [234]. Conversely, our results for miR-125b-1-3p, miR-125b-2-3p and miR-125b-5p showed that they are not significant biomarkers, and miR-125b-5p, for instance, indicated an AUC of 0.4340. In the same meta study, miR-126b-3p and miR-142-3p were listed as a potential biomarkers, and this was confirmed by our results that indicated that miR-126b-3p, the 6th most downregulated miRNA, was a significant potential biomarker, with an AUC of 0.8020. This sort of contradiction illustrates the wide number of different outcomes for the same miRNA in different studies attempting to develop definitive serum-based miRNA biomarkers. We also confirmed that miR-142-3p was a potential biomarker, with an ROC of 0.8696, thus illustrating that multiple miRNAs are potential candidates in our results that have been separately tested in other studies [234]. This is echoed in the literature in which highly comparable miRNA studies with the same objectives yield very different results [235] because multiple variables like age, gender, ethnicity, sample preparation, and detection and quantification of miRNAs can differ across identical studies, thus illustrating the dynamic nature of miRNA expression.

## 4. Materials and Methods

This section outlines the reagents and resources (Table 4), resource availability, the experimental model, the patient details (Table 5), method details and statistical and biostatistical analysis.

### 4.1. Resource Availability

#### 4.1.1. Lead Contact

Further information and requests for resources and reagents should be directed to and will be fulfilled by the Lead Contact.

#### 4.1.2. Material Availability

Serum or RNA generated in this study are available from the Lead Contact with a completed Material Transfer Agreement (some samples are depleted).

#### 4.1.3. Data and Code Availability

The total microRNA-seq datasets illustrating total reads and total UMI per characterized miRNA × 148 sample is currently located in Appendix A.

### 4.2. Experimental Model and Patient Details

#### 4.2.1. Patients and Study Sample

The sample of human subjects involved HCC cases (*n* = 98) that were drawn from two different case control studies, namely, from consenting patients that were diagnosed in the Johannesburg Cancer Case Control Study (JCCCS) [7], a study that recruited self-identified black African (not mixed ancestry) cancer patients in the greater Johannesburg area, Gauteng province, South Africa, and consenting HCC cases that were collected by the oncology departments of Inkosi Albert Luthuli Central Hospital (IALCH) and Greys hospital in Durban and Pietermaritzburg, Kwazulu-Natal, South Africa, respectively. Controls (n-48) included non-HBV infected, otherwise healthy, patients, that were all drawn from the UKZN-based study. Ethics approval for this study was received from the Biomedical Research Ethics Committee (BREC) at the University of KwaZulu-Natal (BREC Reference Number: BE059/15) and the University of the Witwatersrand Human Research Ethics Committee (Medical) (M150239; M140271). All study participants were required to sign an informed consent form before participating in the studies. Patient confidentiality was ensured by recording only case numbers.

#### 4.2.2. Clinical Data Collection

Classification regarding the presence of current HBV infection was not exactly the same in the two case studies. In the JCCCS study, the presence of current infection measured HBV DNA and HBcAb, and its data regarding HBsAg positive and HBeAg positive status was not recorded for many of their HCC cases. Conversely, the UKZN dataset shows HBsAg/HBeAg/HBcAb positive data for every patient but not the presence or absence of HBV-DNA. The two studies thus recorded current HBV infection either by indicating HBsAg or HBV DNA positive status. The evidence of past infection across both studies recorded HBcAb status, as well as well as gender, age and HCV infection. The data indicate that 54% of controls were HBcAb positive and 2% were HBsAg positive (see Table 5). Conversely, the HCC cases indicate that 61% of respondents were either HBsAg or HBV-DNA positive, as well as HBcAb positive, suggesting that a significant majority of HCC patients were either currently infected with hepatitis B or had been exposed to hepatitis B infection that had resolved.

### 4.3. Method Details

#### 4.3.1. Blood Sample Collection

Blood samples were collected from 146 consenting patients using an 18–20 gauge syringe needle and placed in BD Vacutainer serum collection tubes before being centrifuged at 2500× *g* for 20 min at room temperature within 60 min. The serum supernatant was extracted and frozen as 500 uL aliquots and stored at −80 °C.

#### 4.3.2. RNA Extraction

We extracted circulating RNA from 200 μL of serum using the miRNeasy Serum/Plasma Advanced Kit (Qiagen, Redwood City, CA, USA) according to the manufacturer’s protocol. For analysis of the complete plasma RNA, buffer RPL was mixed with the plasma sample to ensure efficient lysis. The kit included buffer RPP, which precipitates contaminants such as plasma proteins, and the kit employed a protocol based on spin column technology. The sample was first applied to the spin column to bind total RNA, and any unwanted analytes or contaminants were removed in subsequent washing and centrifugation steps before the samples were eluted in the final centrifugation step. The HCC serum (200 µL) was mixed with buffer RPL to release and stabilize RNA from plasma proteins and extracellular vesicles. The sample was then mixed with buffer RPP and centrifuged to precipitate proteins. Isopropanol was added to the supernatant to provide the appropriate conditions for RNA molecules (>18 nucleotides) to bind to the silica membrane. The sample was then applied to the RNeasy UCP MinElute spin column, where RNA binds to the membrane and other contaminants are washed away in subsequent wash steps. In the final step, total RNA (>18 nucleotides) was eluted using RNase-free water.

#### 4.3.3. miRNA Sequence

The miRNA sequencing libraries were prepared using the QIAseq miRNA Library Kit. Briefly, adapters were ligated sequentially to the 3′ and 5′ ends of miRNAs before universal cDNA synthesis with UMI assignment. This sequencing, using modified oligonucleotides, eliminated the presence of adapter dimers in the sequencing library and effectively removed a major contaminant often observed during sequencing [236]. The miRNA-seq libraries were amplified by 22 cycles of PCR, during which a unique sample index was added to each sample. The amplified DNA fragments were subjected to double size selection using QIAseq magnetic beads to select the molecules with sizes ranging from 150 to 200 bp. The libraries were quantified using Qubit 3.0 HS dsDNA assay, and the library quality was examined using BioAnalyzer. The miRNA-seq libraries were sequenced at 75 bp single-end reads on an Illumina NovaSeq 6000 (Illumina, San Diego, CA, USA). The 146 samples were sequenced on one Novaseq S2 flowcell with a 100 bp single read run, and one Novaseq SP flowcell with a 100 bp single read run and one 50 bp single read run for barcode QC.

#### 4.3.4. miRNA-seq Data Analysis

All sequencing data were demultiplexed and converted into fastq files by Illumina bcl2fastq (v2.20). The reads were mapped and counted using the primary analysis procedure of Qiagen online miRNA-seq analysis software (https://geneglobe.qiagen.com/sg/analyze, accessed on 10 October 2023) (Version 23.1). Briefly, reads were trimmed off the 3′ adaptor and the low-quality bases and those shorter than 16 bp and less than 10 UMI counts were excluded from the analysis. Alignment was performed using Bowtie with a maximum of two mismatches. Aligned reads were annotated using miRBase (v21). After the removal of duplicates, UMI counts with more than 10 reads were used for miRNA differential expression analysis. UMI read counts for each miRNA was used in the analysis to ensure an accurate reflection of the original miRNA template amount before PCR amplification.

### 4.4. Statistical and Biostatistical Analysis

The mean unique molecular index (UMI) counts and differential miRNA UMI expression for each miRNA for every HCC case (*n* = 98) and control (*n*= 48) was initially calculated using Primary QIAseq miRNA Quantification Data Analysis software (version 2) on the basis of classifying and establishing an UMI count for every classified miRNA nucleotide sequence that has been verified and recorded for every known miRNA [237]. The initial results were assembled on an excel spreadsheet indicating UMI counts for 840 classified miRNAs.

The Ideal R/Bioconductor package for interactive differential expression analysis was deployed to ascertain the differential UMI expression between HCC cases and controls of 840 classified serum-based miRNAs (https://bioconductor.org/packages/ideal/ accessed on 10 October 2023) [238]. The initial count matrix produced by the QIASeq mIRNA Library kit software (Version 2), was normalized before using a negative binomial generalized linear model (GLM) in the R/Bioconductor package that filtered the differential expression of UMI to exclude *p*-value > 0.05. Using a cut-off ≥ 1 for the UMI and an LFC > 2.00 or <−0.50 and with an adjusted *p*-value < 0.05, we identified 148 dysregulated miRNAs, with 49 (33%) upregulated and 99 (67%) downregulated (see Appendix A). A further cut-off of ≥ 10 for the UMI reduced this to 30 upregulated miRNAs and 61 downregulated miRNAs (see Table 2).

Further biostatistical analysis of authenticated miRNA targets and HBV-HCC pathways was investigated by deploying miRPATHv4 (https://diana-lab.e-ce.uth.gr/app/miRPathv4, accessed on 10 October 2023) [239,240].

The identification of potential miRNAs as diagnostic agents was developed from the top 10 most upregulated and downregulated miRNAs (UMI and LFC), as well as from testing additional nine miRNAs that were significantly dysregulated in our results based on their being identified in other studies investigating serum-based miRNAs as diagnostic agents for HBV-HCC detection. A total of 29 potential miRNA candidates were selected, including the top 20 dysregulated miRNAs in terms of UMI and LFC, plus 9 other miRNAs identified in HBV-HCC serum studies as potential biomarkers [71,227,228,230]. These miRNAs were all tested individually for their AUC value using a logistic regression model in which UMI count formed the predictor variable. Each candidate was assessed for its z-score, *p*-value and its relevant AUC score adhering to a *p*-value ≤ 0.05 and an AUC ≥ 0.80. Panels of miRNAs were then assembled to evaluate the one with the highest AUC, sensitivity, specificity and their Youden Index (*p* < 0.05) using Stata 12.0 (http://www.stata.com, accessed on 10 October 2023). A heatmap identifying validated associations with specified signaling pathways, disease and cell processes was developed using miRPATHv4 at https://diana-lab.e-ce.uth.gr/app/miRPathv4, accessed on 10 October 2023) [239,240,241].

## 5. Conclusions and Limitations

This paper provides a comprehensive study of an African miRNome involving dysregulated miRNAs and their targets in HBV-HCC serum drawn from two South African case control studies and, to the best of our knowledge, is the only comprehensive serum-based miRNome study on HCC in Sub-Saharan Africa. In particular, the results focus on the targets and pathways of 30 significantly upregulated miRNAs and 61 downregulated miRNAs (base mean value ≥ 10 UMI, LFC > 2, <−0.5). In addition, the results investigate selected panels of miRNAs as diagnostic biomarkers before selecting the panel with the highest level of specificity and sensitivity. The validated targets of the 91 highly dysregulated miRNAs support the contention that, collectively, miRNAs play a significant role in the modulation of HBV-HCC pathogenesis and that they can be deployed as accurate diagnostic biomarkers.

The primary limitation of this study is that miRNA dysregulation heterogeneity in the face of multiple variables and stages in hepatocarcinogenesis limits the degree to which miRNA results can be generalized. A multitude of diagnostic miRNAs have been developed, for instance, using different miRNAs, even at the same stage of disease. In addition, the usefulness of our diagnostic miRNA panel study is limited because the patient sample involved late-stage HCC rather than early-stage disease. In conclusion, this study acknowledges that despite the undoubted potential of RNA-based therapeutics and diagnostics, this field largely remains a work-in-progress. The problematic nature of miRNA-based therapeutic and diagnostic agents is illustrated by the fact that no results have been published from 34 clinical trials [242]; nevertheless, significant breakthroughs like mRNA-based vaccines illustrate that RNA-based intervention should continue to be pursued. A further limitation is that the patient characteristics across the two case control studies included some different variables like Bilharzia, albumin, ferritin in the UKZN study and not in the JCCCS study, and smoking, drinking, diabetes in the JCCCS study but not in the UKZN study.

Further validation of the gene targets and their effect on gene expression from HCC liver tissue is required for the top dysregulated miRNAs identified in our study, especially miRNAs like let-7a-5p, miR-1246, miR-4532, miR-320b, miR-320c and miR-122-5p.

## Figures and Tables

**Figure 1 ijms-25-00975-f001:**
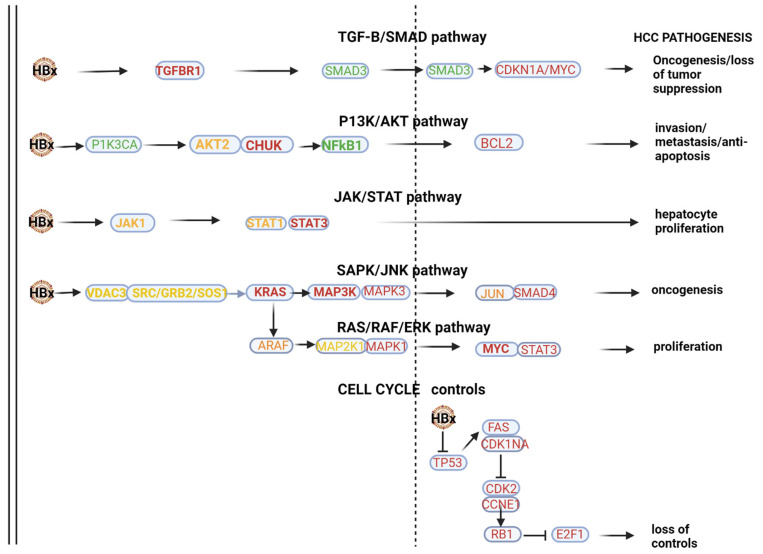
HBV-HCC pathways of top 10 downregulated miRNAs. In the HBx dysregulated HBV-HCC pathways, the top 10 downregulated miRNAs modulate expression in the TGF-B/SMAD pathway by targeting validated gene targets like TGFBR1 and SMAD3/CDKN1A/MYC to influence both oncogenesis and the loss of tumor suppression. In the P13K/AKT pathway, the top 10 downregulated miRNAs target PIK3A and AKT2/CHUK to influence NF-kB1 promotion of BCL2 to influence invasion, metastasis and anti-apoptosis, and in the JAK/STAT pathway, this miRNA panel targets STAT1/3 to influence hepatocyte proliferation. MAPK signaling is targeted by the top 10 downregulated miRNAs by influencing the expression of VDAC3/SRC and GRB2/SOS/KRAS to influence the MAP3K/MAP3/JUN and SMAD4 to influence oncogenesis in the SAPK/JNK. Similarly, MAPK signaling is influenced in the RAS/RAF/ERK pathway by this panel of miRNA targeting ARAF/MAP2KT/MAPK1/MYC and STAT3 to promote cell proliferation. Validated targets in cell cycle controls that are also dysregulated by the HBx protein include TP53/FAS/CDKN1A/CDK2, CCNE1/RBI and E2F1 to influence loss of cell cycle controls. See Table 2 and the Discussion section for further verification of the validated associations in HBV-HCC pathogenesis.

**Figure 2 ijms-25-00975-f002:**
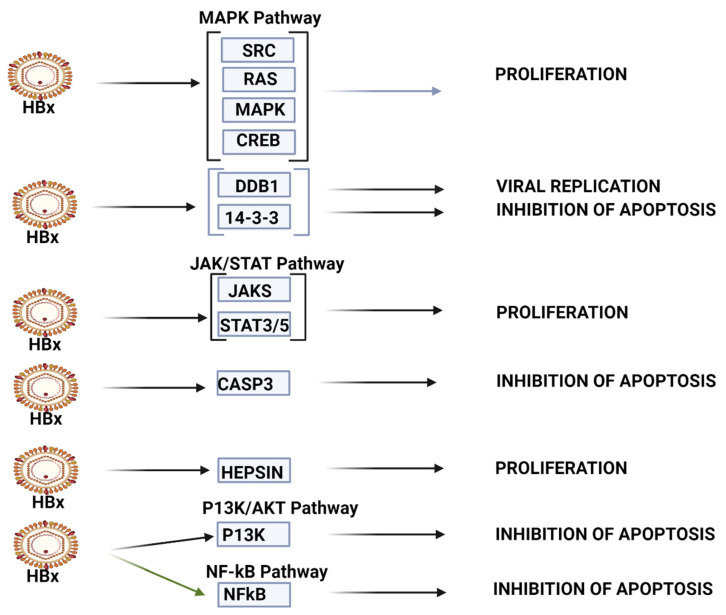
HBV-HCC pathways of top 10 upregulated miRNA. The top 10 upregulated miRNAs influence viral carcinogenesis primarily by modulating cell proliferation and apoptosis. This panel can modulate cell proliferation in MAPK pathways by targeting SRC/RAS/MAPK/CREB, as well as by targeting JAK/STAT3/5 in the JAK/STAT signaling pathway. This panel appears to be able to modulate apoptosis by targeting 14-3-3, CASP3 and p53, as well as P13K in the P13K/AKT pathway and NF-kB in NF-kB signaling (See Table 1 and the Discussion section for further verification of direct HCC role).

**Figure 3 ijms-25-00975-f003:**
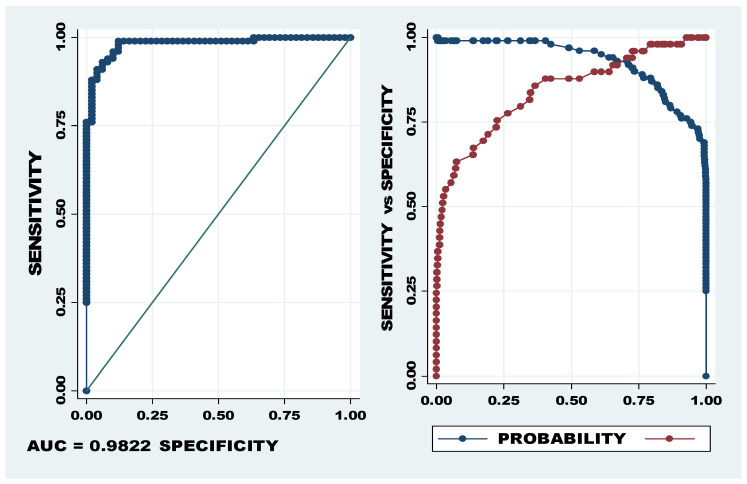
AUC and probability cut-off of let-7a-5p, miR-320b/c. The first graph (LHS) showing sensitivity on the Y axis and specificity on the X axis depicts the AUC of 98.22% and the second graph shows the optimum cut-off point between sensitivity (blue) and specificity (red). The optimum cut-off, indicated as probability cut-off, in the second graph (RHS) indicates a specificity of 95.9% and specificity of 91.0%.

**Figure 4 ijms-25-00975-f004:**
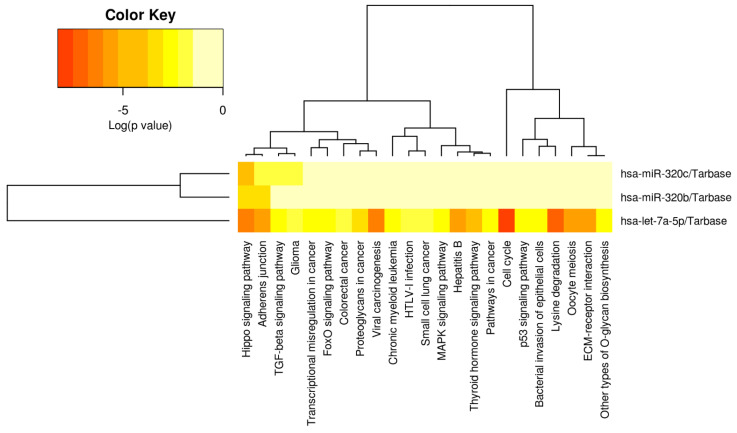
Heatmap of diagnostic miRNA let-7a-5p, miR-320b/c. From left to right, the miRPATHv3 results indicate that let7a-5p is strongly associated with cell cycle, lysine degradation, viral carcinogenesis, the Hippo signaling pathway, Hepatitis B, the Adherens Junction Oocyte meiosis and ECM–receptor interaction (see Table 2/the Discussion section for further validation of direct HCC role). Let-7a-5p has also been associated with the modulation of TGF-B/SMAD signaling, Glioma, transcriptional irregularity in cancer, Fox-O signaling pathway, colorectal cancer, Proteoglycans in cancer, chronic myeloid Leukemia, HTLV-1-infection, small-cell lung cancer, MAPK signaling pathway, p53 signaling, the bacterial invasion of epithelial cells and other types of O-glycan synthesis. The full panel of let-7a-5p, miR-320b/c is only significantly associated with Hippo signaling.

**Table 1 ijms-25-00975-t001:** Top 10 upregulated miRNA UMI ≥ 10.

hsa-miRNA	UMI	LFC	Adj-P	Validated Gene Target	Reference
miR-130b-3p	344	2.579	2.91E−33	upregulated; no validated HCC target	[17]
miR-320d	395	2.508	1.89E−28	Potential Biomarker	[18,19]
miR-483-3p	570.5	3.935	6.31E−37	BRCA1	[20,21]
miR-1246	693.3	4.428	2.46E−53	GSK3β/AXIN2/RORα/CADM1	[22,23,24,25,26]
miR-320b	1244.8	2.018	5.26E−20	PDCD4	[27,28]
miR-192-5p	2452.2	2.537	2.15E−34	XIAP/TRIM44/SEMA3A/FABP3/YY1/PABPC4	[29,30]
miR-4532	3698.6	5.142	5.38E−51	DUSP/PD-L1/MUC1	[27,31]
miR-320c	3978.8	3.708	2.78E−54	GNAI1	[27,32]
miR-483-5p	44,961.1	4.885	1.48E−49	PPARα/TIMP2/CDK15/ALCAM	[21,33,34,35,36]
miR-122-5p	65,595.7	2.135	3.36E−19	β-CAT/CCNG1/p53/HNF4α/ BCL-W/ADAM17	[37,38,39,40,41,42,43,44]

**Table 2 ijms-25-00975-t002:** Top 10 downregulated miRNAs ≥ 10 UMI.

hsa-miRNA	UMI	LFC	Adj-P	Validated HCC Target	Reference
miR-191-5p	2691.1	−0.569	1.3E−08	EGR1/UBE2D3/ZO-1	[45,46]
miR-26b-5p	3497.3	−0.566	1.7E−07	SNAIL/MMP2/SMAD1/KPNA2/PIM-2	[47,48,49,50]
miR-146a-5p	6559.6	−0.904	6.7E−16	CFH/RAC1/ERBB4/IQGAP1/NRA5/PARK2/PTG52	[51,52,53,54]
				GAP1/TRAF6/SMAD4/BRCA1	
miR-142-3p	7768.9	−1.274	9E−18	LDHA/HMGB1/RAC1/ZEB1/CD133/SLC3A2	[51,52,53,54]
miR-126-3p	9528.3	−0.608	1.3E−07	LRP6/PIK3R2GAS5/SPRED1/ADAM9	[55,56,57,58]
let-7i-5p	9798.9	−0.922	3.2E−15	TSP1/CD47	[59]
let-7f-5p	13,136.2	−1.236	1.2E−19	No direct HCC reference	n/a
let-7a-5p	13,460.6	−1.155	9E−18	MMP11/BZW2//STAT3/IAP3/HMGA2/IGFBP2/3	[60,61,62,63,64]
let-7b-5p	17,716.1	−0.889	7.9E−08	CDC25B/HMGA2/GPC3	[65,66]
miR-16-5p	57,609.4	−0.543	<0.001	GF1R/ANXA11/IGF1R	[67,68,69,70,71]

**Table 3 ijms-25-00975-t003:** Potential diagnostic miRNA panels.

miRNA	*p*-Value	R^2^	AUC
Combination 1			0.9708
Let-7a-5p	<0.001	0.6684	
miR-1246	<0.001		
Combination 2			0.9363
let-7f-5p	<0.001	0.5375	
miR-4532	<0.001		
Combination 3			0.9737
miR-320c	<0.001	0.656	
Let-7a-5p	<0.001		
Combination 4			0.9420
miR-122-5p	<0.001	0.4656	
Let-7a-5p	<0.001		
Combination 5			0.9822
miR-320b	0.004	0.7347	
miR-320c	<0.001		
Let-7a-5p	<0.001		

**Table 4 ijms-25-00975-t004:** Resources and reagents used for methods.

Reagent or Resources	Source	Identifier
**Biological Sample** Serum from HCC patients		
**Commercial assay kits** miRNeasy Serum/Plasma Advanced Kit (50); For 50 total RNA preps: 50 RNeasy UCP MinElute Spin Columns, Collection Tubes (1.5 mL and 2 mL), RNase-free Reagents and buffers	Qiagen	217204
Qiaseq miRNA library kit (12); For 12 sequencing prep reactions: 3′ ligation, 5′ ligation, reverse transcription, cDNA cleanup, library amplification and library cleanup reagents; quality control	Qiagen	331502
**Deposited data** Characterization miRNAs (*p* < 0.05) that were deposited in this paper (see Appendix A)	This paper	
**Software and algorithms** miRBasev21; Ideal R/Bioconductor; miRPathv4; miRTargetLink2	Internet	https://www.mirbase.org (version 21, accessed on 10 October 2023) http://bioconductor.org/packages/ideal (version 3.18, accessed on 10 October 2023) https://diana-lab.e-ce.uth.gr/app/miRPathv4 (version 4, accessed on 10 October 2023) https://ccb-compute.cs.uni-saarland.de (version 2.0, accessed on 10 October 2023)

**Table 5 ijms-25-00975-t005:** Sample characteristics.

Cases	Gender	Age (yrs.)	HCV+	HBsAg/HBV-DNA	HBcAb
**Controls (48)**	Male 62.5%	42.75 (12.1)	0%	2%+	54%+
**HCC cases (98)**	Male 70.4%	44.0 (13.2)	7.15%+	61.2%+	61.2%+

## Data Availability

All of the data files generated by the differential analysis software including UMI expression of 840 classified miRNAs in both cases and samples are available on request.

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
