# Peer review of "Serum microRNA Profiles and Pathways in Hepatitis B-Associated Hepatocellular Carcinoma: A South African Study"

_ijms, 2024, doi:10.3390/ijms25020975_

Round 1
Reviewer 1 Report
Comments and Suggestions for Authors
The manuscript "Serum microRNA profiles and pathways in hepatitis B-associated hepatocellular carcinoma: A South African study" is very interesting and contributes to the field. The methods and data presented are adequate and comprehensive. However, the main problem with this manuscript is the figures. Authors are asked to revise the figures by increasing the resolution and making the pathway easier for readers to understand.
Comments on the Quality of English LanguageEnglish needs to be checked for grammar.
Author Response
Dear Reviewer 1 please see attached response.
Many thanks, your review added considerable value.
best wishes
KS et al

Reviewer 2 Report
Comments and Suggestions for Authors
The manuscript focuses on the profiling of serum miRNA and understanding the metabolic pathways involved in hepatitis B-associated hepatocellular carcinoma in South African subjects. The idea of the manuscript is original and the findings can support current studies on cancer by providing associated molecular mechanisms. I have several notes for improvement:
1. Results section: please check the numbering of the tables. Why Table 2 and 3 are presented in the text before Table 1?
2. Figure 3 and 4: please provide clearer figures with a better quality.
3. I found several unnecessary spacing between words. In some cases, spacinga are also missed. Please recheck your whole manuscript to address these issues.
4. Reference 355 is presented in the text but not in the references list section.
5. Could you please check your manner of citation? Because I found the citation numbering was quite confusing. Did you number your references based on the order they appeared in your text? Please recheck and addredd this issue.
6. If possible, please provide comparison between your study (Sub-Saharan and South African subjects) with other similar studies representing other communities.
Thank you and good luck.
Author Response
Dear Reviewer
Many thanks for your input. We have tried to the best of our ability to make all the necessary changes suggested. Please see point by point response.
Sincerely
Kurt Sartorius et al

Reviewer 3 Report
Comments and Suggestions for Authors
Hepatocellular carcinoma (HCC) ranks as the fifth most common cancer, exhibiting high mortality rates primarily due to late-stage diagnoses and limited treatment options. Serum miRNAs have emerged as potential novel biomarkers for the early detection of HCC. Sartorius et al. conducted miRNome profiling to identify dysregulated miRNAs between HCC and normal healthy volunteers. They identified 91 dysregulated miRNAs, comprising 30 upregulated and 61 downregulated ones. Based on these findings, they developed a diagnostic miRNA panel. While these results could provide valuable insights into diagnostic and therapeutic approaches for chronic HBV patients, it's essential to note that some statements in this manuscript lack sufficient support from the presented data and may necessitate additional experiments to validate the conclusions.
Specific point
1. In the final sentences of the Introduction section, the authors described 'The introduction should briefly place the study in a broad context and highlight why it is important. It should define the purpose of the work and its significance. The current state of the research field should be carefully reviewed and key publications cited. Please highlight controversial and diverging hypotheses when necessary. Finally, briefly mention the main aim of the work and highlight the principal conclusions. As far as possible, please keep the introduction comprehensible to scientists outside your particular field of research. References should be numbered in order of appearance and indicated by a numeral or numerals in square brackets—e.g., [1] or [2,3], or [4–6]. See the end of the document for further details on references.' Notably, this text seems to be part of the MDPI template (https://www.mdpi.com/files/word-templates/symmetry-template.dot). It raises the question of whether the authors thoroughly reviewed the final manuscript before submitting it.
2. The section number of Materials and Methods is specified as 5. However, in this section, subnumbers are assigned starting from 2.1. Please make the necessary correction to ensure consistency in the numbering.
3. Please correct "HBcAB" to "HBcAb" and "HBsAG" to "HBsAg" for accuracy.
4. In section 2.2.3 concerning clinical data collection and Table 1, the authors categorized patients into two groups (control, n=48, and HCC, n=98). However, given that the primary focus of this paper is the involvement of serum microRNA in hepatitis-B-associated hepatocellular carcinoma, the inclusion of 2% of HBsAg/HBV DNA-positive individuals in the control group may not be suitable in the context of this study. Additionally, it is recommended to provide patient information such as Edmondson grades or HCC stages, particularly for patients diagnosed with HCC. If feasible, please also include details on amount of HBVsAg or HBV-DNA, AFP levels and treatment modalities for a more comprehensive understanding.
5. In Result, the authors described ‘Of these, 17/31 upregulated miRNA exceeded LFC > 3 sug-gesting major changes in expression between the controls and the HCC cases in terms of both UMI and LFC.’. Please confirm whether it should be "17/30" or "17/31," Please verify the data and statement in your original text to ensure accuracy.
6. In the legend of Figure 1, the author mentioned, ‘ Figure 1. HBV-HCC pathways of top 10 downregulated miRNA. Figure 1. HBV-HCC pathways of top 10 downregulated miRNA. Figure 1: HBV-HCC pathways of top 10 downregulated miRNA (gene targets highlighted in yellow, amber and red indicate ascending level of validation in terms of HCC studies-miRPATHv4)?’. Please review and make any necessary corrections for clarity and consistency.
7. Figure 1 visualizes the HBV-HCC pathway involving downregulated miRNA using a pathway diagram, but it appears very busy and challenging to comprehend. Would it be better to narrow down the selected miRNA candidates and simplify the figure?"
8. In this study, the authors analyze the variations in miRNA present in the serum of HCC patients through miRNA-seq, comparing them with those of healthy individuals. To enhance the value of the obtained data, considering the ambiguity in patient information, it would be beneficial to investigate the changes at the mRNA or protein level of the genes illustrated in Figures 1 and 2 (at least, target for Let-7a-5p, miR-1246, miR-4532, miR-320b, miR-320c and miR-122-5p) in biopsy liver samples from HCC patients.
9. The text in Figures 3 and 4 is unreadable. Please correct.
10. LHS and RHS should be corrected.
11. Please discuss the results obtained in this study, making comparisons with findings from previous reports (e.g., PMID: 37383532, PMID: 35731638, PMID: 36457743, PMID: 22174818, PMID: 28079796, and PMID: 31320713).
Comments on the Quality of English LanguageThe authors should carefully review the final manuscript before submission, as there are several mistakes and sentences that are not related to the content in this manuscript.
Author Response
Dear Reviewer
Thank you for your detailed review. We have made major changes and attempted to deal with all of your queries
Sincerely
Kurt Sartorius et al

Round 2
Reviewer 1 Report
Comments and Suggestions for Authors
The revision is good!
Author Response
Dear Reviewer
Once again thank you for your contribution.
Best wishes
kurt sartorius et al
Reviewer 3 Report
Comments and Suggestions for Authors
-
1. The content of Figure 2 and Figure 3 is acceptable, but the quality of these figures is very low. Please adjust the font and size of the text to match the quality of the paper. The current appearance with crushed and unattractive text is undesirable.
-
2. Figure 4 is difficult to interpret due to unclear color keys. Please make corrections to enhance the clarity of the color key.
Acceptable
Author Response
Dear Reviewer
I have redone Figure 2 using Biorender, I think I should have done this in the first place but it looks much better. I have attempted to improve clarity of Figure 3 so text is not crushed. Have also sharpened up color chart in figure 4. The diagrams coming out of miRPATH quite difficult to edit especially text, clarity. I will also confer with IJMS re nicest version of Figs 3-4.
Once again thank you for your input
Best wishes
Kurt Sartorius et al